# Stochastic force inference via density estimation

**Victor Chardès**[1,†], **Suryanarayana Maddu**[1,†], **Michael J. Shelley**[1,2]

[1] Center for Computational Biology,
Flatiron Institute, New York, NY, USA, 10010
[2] Courant Institute of Mathematical Sciences,
New York University, New York, NY, USA, 10012

## Abstract

Inferring dynamical models from low-resolution temporal data continues to be a significant challenge in biophysics, especially within transcriptomics where separating molecular programs from noise remains an important open problem. We explore a common scenario in which we have access to an adequate amount of cross-sectional samples at a few time-points, and assume that our samples are generated from a latent diffusion process. We propose an approach that relies on the probability flow associated with an underlying diffusion process to infer an autonomous, nonlinear force field interpolating between the distributions. Given a prior on the noise model, we employ score-matching to differentiate the force field from the intrinsic noise. Using relevant biophysical examples, we demonstrate that our approach can extract non-conservative forces from non-stationary data, that it learns equilibrium dynamics when applied to steady-state data, and that it can do so with both additive and multiplicative noise models.

## 1 Introduction

**Learning dynamical models**    From gene expression in cells [1–3] to collective motion in animal groups [4] to growth in ecological communities [5], biological processes undergo stochastic dynamics, and their steady-states emerge from a competition between intrinsic noise and deterministic forces. The ability to separate these two contributions from experimental data is crucial to understanding such dynamics. Instrumentally, such systems can be modeled as diffusion processes for which the time-continuous evolution of the degree of freedom of interest $\mathbf{x} \in \mathbb{R}^d$ obeys an autonomous stochastic differential equation [6]:

$$d\mathbf{x} = \mathbf{f}(\mathbf{x})dt + \sqrt{2}\mathbf{G}(\mathbf{x})d\mathbf{W}, \tag{1}$$

where $\mathbf{W}$ is a standard $d$-dimensional Wiener process, $\mathbf{f} : \mathbb{R}^d \to \mathbb{R}^d$ is the deterministic force, $\mathbf{G} : \mathbb{R}^d \to \mathbb{R}^{d \times d}$ the diffusion coefficient and the equation is written in the Itô convention. This formulation in terms of stochastic trajectories $\{\mathbf{x}(t), t > 0\}$ is equivalent to a formulation in terms of the probability density $p_t(\mathbf{x})$ given by the Fokker-Planck equation:

$$\partial_t p_t(\mathbf{x}) = -\nabla \cdot [\mathbf{f}(\mathbf{x})p_t(\mathbf{x}) - \nabla \cdot (\mathbf{D}(\mathbf{x})p_t(\mathbf{x}))], \tag{2}$$

where $\mathbf{D} = \mathbf{G}\mathbf{G}^T$ and the divergence is applied row-wise for any matrix-valued function. Within this framework, the inverse problem of interest is to estimate $\mathbf{f}(\mathbf{x})$ from experimental data given the knowledge of the diffusion field $\mathbf{D}(\mathbf{x})$. In soft-matter and finance, due to the availability of time-resolved trajectories, this problem has largely been addressed through discretizations of (1) [7–10]. For the higher dimensional systems encountered when studying gene expression, single-cell sequencing technologies give access to many cross-sectional samples with low temporal resolution.

---

[†]These authors contributed equally

NeurIPS 2023 AI for Science Workshop.

In this scenario, the inverse problem amounts to learning how the probability mass is moved between empirical distributions at successive time points rather than how one trajectory evolves over time. Previous attempts to tackle this question have approached it from an optimal transport point of view; first in a static setting by learning pairwise couplings between successive empirical distributions, and subsequently in a dynamical setting by learning a time-continuous model connecting distributions at all times. While static methods couldn't model time-continuous and non-linear dynamics [11–15], their dynamical counterparts lifted these constraints but remained limited to diffusion processes with additive noise [16–19]. The ability of multiplicative noise to move, create, and destroy fixed points in the energy landscape, in particular in the context of biochemical networks [2, 3, 20], motivates the development of an inference scheme able to handle it.

**Equilibrium vs. non-equilibrium**    Biological systems exhibit a distinctive feature: they operate out of equilibrium at the molecular level. For instance, irreversible chemical cycles orchestrate various molecular processes in cells, ranging from the work of molecular motors to phosphorylation cycles, enzymatic cycles, and RNA transcription regulation [21, 22]. The dissipation of chemical energy within these cycles induces irreversible transitions between distinct molecular states, maintaining biological systems far from thermodynamic equilibrium. Learning these irreversible cycles is essential for understanding and predicting the behavior of such processes, but it remains a challenge when the time resolution of experiments is limited. To illustrate this issue in the framework of diffusion processes, we consider the problem of estimating the force field $\mathbf{f}(\mathbf{x})$ knowing the noise $\mathbf{D}(\mathbf{x})$ and using a large number of samples drawn from the steady-state distribution $p(\mathbf{x})$ of (2). At steady-state, the Fokker-Planck equation reduces to:

$$\nabla \cdot \mathbf{j}(\mathbf{x}) = 0, \tag{3}$$
$$\mathbf{j}(\mathbf{x}) = (\mathbf{f}(\mathbf{x}) - \nabla \cdot \mathbf{D}(\mathbf{x}) - \mathbf{D}(\mathbf{x})\nabla \log p(\mathbf{x}))\, p(\mathbf{x}),$$

where we assume that $\mathbf{D}(\mathbf{x})$ is invertible and we denote $\mathbf{j}(\mathbf{x})$ the probability current. An estimate of the score function $\mathbf{s}(\mathbf{x}) \simeq \nabla \log p(\mathbf{x})$ which does not require computation of the partition function [23], provides access to a force field $\mathbf{f}^{\mathrm{eq}}$ describing an equilibrium steady-state (with vanishing currents):

$$\mathbf{f}^{\mathrm{eq}}(\mathbf{x}) = \mathbf{D}(\mathbf{x})\mathbf{s}(\mathbf{x}) + \nabla \cdot \mathbf{D}(\mathbf{x}) \simeq \mathbf{f}(\mathbf{x}) - \mathbf{j}(\mathbf{x})/p(\mathbf{x}). \tag{4}$$

This relation was exploited in the recent study [24] to learn a coarse-grained (CG) force field in molecular dynamics using an estimate of the score function $\mathbf{s}(\mathbf{x})$. To infer $\mathbf{f}(\mathbf{x})$ from (4), one needs to estimate the full probability density of the model $p(\mathbf{x})$ as well as the probability currents $\mathbf{j}(\mathbf{x})$. However, at steady-state and with only the knowledge of $p(\mathbf{x})$ and its score there is no constraint, besides being divergence-free, on $\mathbf{j}(\mathbf{x})$. This degeneracy is lifted when trajectory information is available [25, 26], but in the absence of this information non-equilibrium currents are only accessible using data sampled from a non-stationary solution of (2). This issue was observed in early attempts to learn dynamical models from single-cell RNA-seq data [20, 27, 28].

**Our contribution**    In this paper, we develop an inference framework to estimate the force field $\mathbf{f}(\mathbf{x})$ from limited static cross-sectional measurements over time, as illustrated in Fig. 1. Our dataset consists of empirical distributions $(\hat{\nu}^1, ..., \hat{\nu}^K)$ known at successive times $t_1 < ... < t_K$, where:

$$\hat{\nu}^k(\mathbf{x}) \stackrel{\text{def.}}{=} \frac{1}{N_k} \sum_{i=1}^{N_k} \delta(\mathbf{x} - \mathbf{x}_i(t_k)), \tag{5}$$

with $N_k$ the number of measurements available from the marginal $p_{t_k}(\mathbf{x})$ at time $t_k$. Our approach does not presuppose that successive samples form a trajectory of the diffusion process (1), but it reduces to least-square-based force inference methods (akin to [8]) if trajectory information is available. We propose a method that simultaneously allows us to (i) learn a continuous-time dynamical model with an autonomous non-linear force field and (ii) separate the force field from a known arbitrary noise model. We show that by using non-stationary data our method can learn non-conservative forces and that it learns equilibrium dynamics when applied to steady-state data. Furthermore, we show that our method can extract force fields from both additive and multiplicative noise models, opening promising applications for gene regulatory network inference.

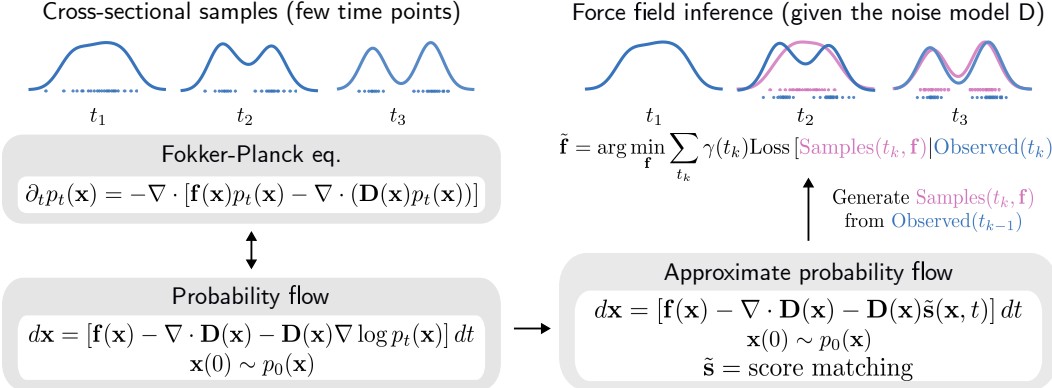

Figure 1: **Force field inference of a diffusion process given cross-sectional samples at a limited number of time points.** We assume that the samples are generated through an underlying diffusion process with an unknown force field $\mathbf{f}(\mathbf{x})$ and a known noise model $\mathbf{D}(\mathbf{x})$. We then define a corresponding approximate probability flow ODE by substituting the score by its continuous-time analog $\tilde{\mathbf{s}}(\mathbf{x}, t)$ estimated via score-matching. Using this approximate probability flow ODE we solve a density fitting problem to learn a force field that best reconstructs the cross-sectional measurements.

## 2 Related Work

The force inference problem can be conveniently reformulated as a *trajectory inference* exercise where the goal is to infer the latent dynamics tracing the cross-sectional measurements. In this section, we briefly outline related works in the context of trajectory inference.

**Deterministic transport** Most approaches proposed to solve the problem of *trajectory inference* have taken an optimal transport (OT) route: recovering latent dynamics that obey the principle of least-effort. These approaches learn a deterministic map that solves a Monge-Kantorovich problem between each pair of successive snapshots [11–13] resulting in a time-discontinuous solution. This constraint was later relaxed using a neural ODE [29] approximating the Benamou-Brenier formulation of OT as regularized continuous normalizing flow [16]. Although OT-inspired transport flow maps are very attractive, they restrict the force fields to be curl-free and are thereby constrained to learn only equilibrium dynamics.

**Stochastic transport with additive noise** Although successfully applied to data [12, 15], OT approaches also can't infer or accommodate stochastic dynamics. In this direction, the Schrodinger bridge (SB) framework aims to find the most likely diffusion process with additive noise connecting a pair of successive snapshots. The SB formulation can be mapped on an entropically-regularized optimal transport problem [14, 19], and therefore inherits the limitation of the solution being time-discontinuous. The studies [17, 18] tackle this issue by constructing a constant estimator for the distribution over trajectories of a diffusion process with constant additive noise.

**Towards general noise models** Despite the success of SB-inspired methods, they are usually limited to additive noise models. Approaches based on Neural SDEs are less constrained than SB-inspired methods and offer flexibility to handle arbitrary noise models. However, they suffer from issues related to robustness and overfitting [30] originating from the need to solve SDEs during optimization. In this paper, leveraging the probability flow ODE [31] and score matching methods [23, 32], we reformulate the neural SDE approach of [28] into a neural ODE, and show its ability to extract a non-conservative force fields from arbitrary noise models.

## 3 Methods

**Probability flow ODE** We consider the SDE in (1), for which the probability density $p_t(\mathbf{x})$ evolves according to the Fokker-Planck equation (2). It can be shown [31, 33] that there exists a corresponding deterministic process whose trajectories share the same marginal probability density as the SDE

described in (1).

$$dx = [\mathbf{f}(\mathbf{x}) - \nabla \cdot \mathbf{D}(\mathbf{x}) - \mathbf{D}(\mathbf{x})\nabla \log p_t(\mathbf{x})] \, dt, \text{ with } \mathbf{x}(0) \sim p_0(\mathbf{x}), \tag{6}$$

where $p_0(\mathbf{x})$ is the distribution at initial time. With knowledge of the noise model $\mathbf{D}(\mathbf{x})$ and an autonomous force field $\mathbf{f}(\mathbf{x})$, the solution path of the ODE (6) connects the empirical distributions $(\hat{\nu}^1, ..., \hat{\nu}^K)$. However, this requires an estimate of the score $\nabla \log p_t(\mathbf{x})$ that can be computed from the cross-sectional samples at the observed time points.

**Score estimation**   We use score matching [23] to estimate the score using samples from the cross-sectional measurements. It amounts to solving the following optimization problem:

$$\tilde{\mathbf{s}} = \arg\min_{\mathbf{s}} \sum_{k=1}^{K} \lambda(t_k) \mathbb{E}_{p_{t_k}} \left[ \text{tr}\left(\nabla \mathbf{s}(\mathbf{x}, t_k)\right) + \frac{1}{2} \|\mathbf{s}(\mathbf{x}, t_k)\|_2^2 \right], \tag{7}$$

where $\lambda : [t_1, t_K] \to \mathbb{R}^+$ is a positive weighting function. We parameterize the score $\mathbf{s}(\mathbf{x}, t) : \mathbb{R}^d \to \mathbb{R}^d$ using a fully connected neural network, and use sliced score matching [32] to estimate score in high dimensions. During training the weights $\lambda$ are automatically tuned based on the variance normalizing strategy proposed in [34]. The time-continuous score $\tilde{\mathbf{s}}(\mathbf{x}, t)$, estimated by solving the optimization problem in (7), is used to define the approximate probability flow ODE:

$$dx = [\mathbf{f}(\mathbf{x}) - \nabla \cdot \mathbf{D}(\mathbf{x}) - \mathbf{D}(\mathbf{x})\tilde{\mathbf{s}}(\mathbf{x}, t)]dt. \tag{8}$$

Using this ODE we can now learn the force field that best reconstructs the cross-sectional measurements $(\hat{\nu}^1, ..., \hat{\nu}^K)$ given the noise model $\mathbf{D}(\mathbf{x})$.

**Force field inference via density fitting**   Using the approximate probability flow ODE (8) and the force field $\mathbf{f}(\mathbf{x})$, one can evolve samples $\{\mathbf{x}_i(t_k), 1 \leq i \leq N_k\}$ to any future time. Our inference task therefore reduces to a density fitting problem that learns a force field $\tilde{\mathbf{f}}_\theta(\mathbf{x})$ connecting through the probability flow ODE the empirical distributions at successive time points:

$$\hat{\theta} = \arg\min_{\theta} \sum_{k=1}^{K} \gamma(t_k) \mathcal{L}(\hat{\mu}_\theta^k, \hat{\nu}^k), \text{ with } \hat{\mu}_\theta^k \stackrel{\text{def.}}{=} \frac{1}{N_{k-1}} \sum_{i=1}^{N_{k-1}} \delta(\mathbf{x} - \tilde{\mathbf{x}}_i(t_k)) \tag{9}$$

$$\text{s.t. } \tilde{\mathbf{x}}_i(t_k) = \mathbf{x}_i(t_{k-1}) + \int_{t_{k-1}}^{t_k} \left( \mathbf{f}_\theta(\mathbf{x}_i) - \nabla \cdot \mathbf{D}(\mathbf{x}_i) - \mathbf{D}(\mathbf{x}_i)\tilde{\mathbf{s}}(\mathbf{x}_i, \tau) \right) d\tau \tag{10}$$

$$\forall i \in \{1, ..., N_{k-1}\},$$

where $\mathcal{L}$ is a loss function that minimizes the distance between $\mu_\theta$ and the empirical measure $\nu$, and $\gamma$ is a positive weighting function. Although there exist several approaches to compute the distance between measures like Maximum Mean Discrepancy (MMD), or [35], Kullback-Leibler divergence, we rely on the Sinkhorn divergence [36] for the density fitting problem. The latter approach leverages the geometry of Optimal Transport and the favorable high-dimensional sample complexity of MMD. We also note that, if the particle trajectories are known, the loss $\mathcal{L}$ reduces to computing the Euclidean distance and the inference problem is similar to least-square-based force inference methods [8].

**Training**   Both the score $\mathbf{s}(\mathbf{x}, t)$ and the non-linear field $\mathbf{f}(\mathbf{x})$ are parameterized by fully connected neural networks with Sine activations [37]. The score network takes as input the sample $\mathbf{x}$ and time $t$ and outputs the time-continuous score estimate, whereas the force field network takes as input only the sample $\mathbf{x}$. In the density fitting problem described in (9), we discretize the probability flow ODE using the explicit Euler method during training and backpropagate gradients through the solver steps. We use Adam Optimizer to train our neural networks.

## 4   Results

To benchmark our force field inference approach we tested two diffusion models: one with additive noise, non-reciprocal linear interactions, and anisotropic diffusion, and one chemical reaction system exhibiting bistability driven by multiplicative noise. To generate samples from these diffusion processes we discretize the SDEs in (1) using the Euler-Maruyama scheme:

$$\mathbf{x}(t + \delta t) = \mathbf{x}(t) + \delta t \, \mathbf{f}(\mathbf{x}(t)) + \sqrt{2\delta t \overline{\mathbf{D}(\mathbf{x})}}\xi, \tag{11}$$

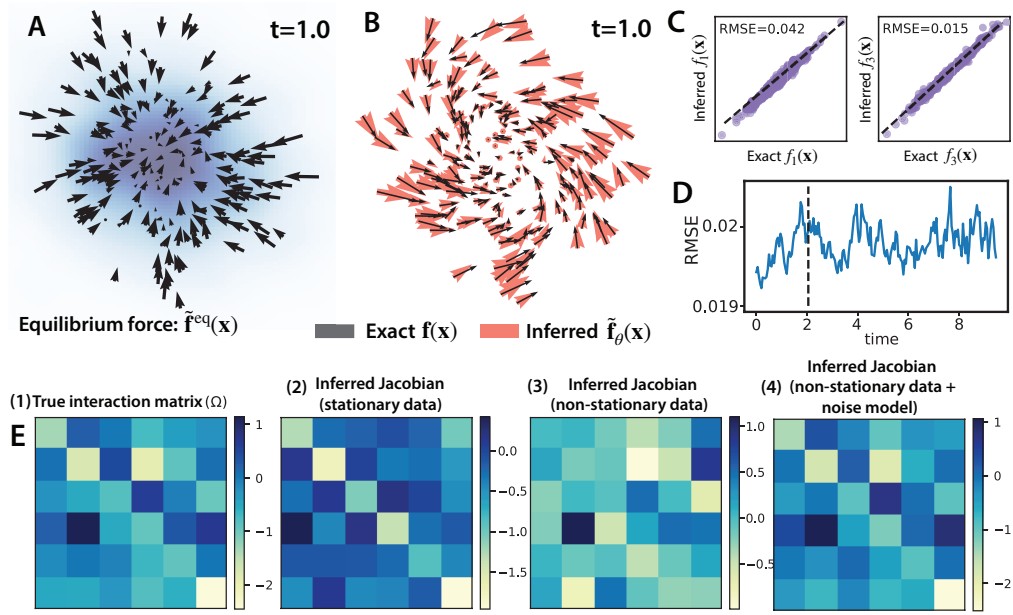

Figure 2: **Non-equilibrium 6D Ornstein-Uhlenbeck process.** (A) Equilibrium force $\tilde{\mathbf{f}}^{\mathrm{eq}}(\mathbf{x})$ inferred from stationary distribution. The background watercolor corresponds to the density field estimated from the samples. (B) The inferred force field (red arrows) overlaid on the exact force field (black arrows). (C) Inferred force field vs. exact force field (1st and 3rd components). (D) Relative Mean-Squared Error $(\mathrm{RMSE}) = \|\tilde{\mathbf{f}}_\theta - \mathbf{f}\|_2^2 / \|\mathbf{f}\|_2^2$ as a function of time. The force field is inferred from cross-sectional samples taken before a specific point in time, indicated by a dashed line. (E) Comparison of interaction matrices obtained from inference with stationary data (E2), non-stationary data (E3), non-stationary data + noise model (E4), with the interaction matrix $\mathbf{\Omega}$ of the true process (E1).

where $\xi$ is a vector of independent normal random variables with zero mean and unit variance. We run multiple realizations of the discretized SDE with time-step $\delta t$ to generate empirical distributions at a given time.

**Non-equilibrium Ornstein-Uhlenbeck process**   We consider a 6-dimensional Ornstein-Uhlenbeck process with diffusion and interaction terms chosen to break the detailed balance and create irreversible phase-space currents at steady state [6]:

$$d\mathbf{x} = -\mathbf{\Omega}\mathbf{x}dt + \sqrt{2\mathbf{D}}d\mathbf{W}, \tag{12}$$

where $\mathbf{\Omega}$ is the interaction matrix, $\mathbf{D}$ the diffusion term (symmetric positive semi-definite matrix) and $\mathbf{W}$ a standard 6-dimensional Wiener process. We consider the non-symmetric interaction matrix and the anisotropic diffusion matrix used as a benchmark for a trajectory-based force inference method in [8]. We run 5000 realizations of the process to generate empirical distributions with $\delta t = 0.01$ and for inference, we use distributions sampled at $\Delta t = 0.1$ with $K = 20$. First, we estimate the score via sliced score matching [32] using a neural network with 3 hidden layers containing 20 nodes each. The force field network is modeled using a neural network with 2 hidden layers containing 10 nodes per layer. In Fig. 2, we illustrate how our framework enables accurate reconstruction of the force field from a few time snapshots of the empirical distributions, and in particular how it captures the non-reciprocal interactions, which manifest themselves as spirals in the 2-dimensional projections of Fig 2B. On the contrary, we see in Fig 2A that our approach captures just reciprocal (symmetric) interactions when applied to stationary data and learns a radial vector field instead. Given the autonomous nature of the inferred non-linear field, we can predict beyond the training horizon as shown in Fig.2D.

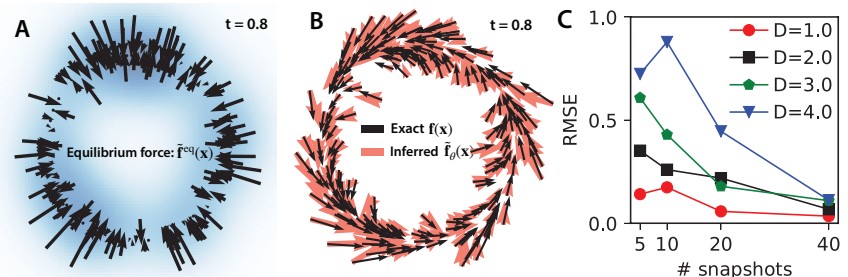

Figure 3: **Non-equilibrium 2D Ornstein-Uhlenbeck process whith harmonic trapping at the origin where $\mathbf{f}(\mathbf{x}) = -\mathbf{\Omega}\mathbf{x} + \alpha e^{-x^2/2\sigma^2}\mathbf{x}$, where $\mathbf{\Omega} = \left(\begin{smallmatrix} 2 & 2 \\ -2 & 2 \end{smallmatrix}\right)$.** (A) Equilibrium force $\tilde{\mathbf{f}}^{\mathrm{eq}}(\mathbf{x})$ inferred from stationary data. (B) The inferred force field (red arrows) overlaid on the exact force field (black arrows). (C) RMSE versus sampling rate $\Delta t \sim 1/N_k$ for different diffusion coefficients. We ran 1000 realization of the process with $\delta t = 0.001$ as per the discretization in (11) with $\alpha = 10, \sigma = 2$, and sample empirical distributions with $\Delta t = \{0.025, 0.05, 0.1, 0.2\}$ during inference. The score was estimated with a neural network with 4 hidden layers containing 10 nodes each. The force field is inferred via a neural network with 2 hidden layers and 10 nodes per layer.

Once the force field is inferred, we can then compute the interaction matrix as the Jacobian of the estimated force field:

$$\tilde{\mathbf{J}}(\mathbf{x}) = \left( \frac{\partial \tilde{\mathbf{f}}_\theta}{\partial x_1} \quad \dots \quad \frac{\partial \tilde{\mathbf{f}}_\theta}{\partial x_d} \right), \quad \widetilde{\mathbf{\Omega}} \simeq \frac{1}{N_k} \sum_{k=1}^{K} \langle \tilde{\mathbf{J}}(\mathbf{x}(t_k)) \rangle,$$

where $\langle ... \rangle$ represents the average over all measured cross-sectional samples. Since the force field is parameterized by a neural network, we can use backpropagation to evaluate the Jacobian. We see using (9) that at steady-state the equilibrium interaction matrix recovered $\widetilde{\mathbf{\Omega}}^{\mathrm{eq}}$ reads:

$$\widetilde{\mathbf{\Omega}}^{\mathrm{eq}} \simeq \mathbf{D}\nabla^2 \log p(\mathbf{x}). \tag{13}$$

From this equation, we can see that in the presence of anisotropic diffusion, the equilibrium interaction matrix is non-symmetric. In Fig. 2E2 we illustrate this fact as the interaction matrix learned on stationary data (and thus corresponding to vanishing phase-currents) is non-symmetric. Importantly, and in agreement with the results shown Fig. 2A-B, the estimated equilibrium matrix $\widetilde{\mathbf{\Omega}}^{\mathrm{eq}}$ is an inaccurate reconstruction of the true interaction matrix $\mathbf{\Omega}$ as shown in Fig. 2E1 and 2E2. Following this observation, we show in Fig. 2E3 that inferring on non-stationary data without noise prior ($\mathbf{D} = 0$, akin to TrajectoryNet [16]) does not permit a faithful reconstruction of $\mathbf{\Omega}$, and that only our complete inference scheme in Fig. 2E4 can accurately infer the interaction matrix.

We evaluate our method's effectiveness in inferring non-linear fields in a 2D OU process with harmonic trapping. In Fig. 4, we show the reconstructed force field and also investigate the impact of sampling rate and diffusion constant on inference accuracy, noting a decline in accuracy with increased diffusion coefficients and reduced sampling rates.

**Stochastic chemical kinetics**   Multiplicative noise is ubiquitous in biological processes whose properties emerge from birth-death dynamics [6]. To illustrate the ability of our method to deal with this type of system, we consider the Schlögl model, a canonical chemical reaction system that exhibits bistability:

$$A + 2X \underset{k_2}{\overset{k_1}{\rightleftharpoons}} 3X, \ B \underset{k_4}{\overset{k_3}{\rightleftharpoons}} X, \tag{14}$$

where species $A$ and $B$ are kept at a constant concentration, denoted $a$ and $b$ respectively. We can describe the stochastic evolution of this process using a Chemical Master Equation (CME) [6], which extends the deterministic law of mass-action to account for fluctuations in copy numbers of each species. Denoting $x$ the concentration of species $X$, the CME can be approximated by a Chemical Langevin Equation (CLE) [38]:

$$dx = [u(x) - v(x)] \, dt + \frac{1}{\sqrt{V}} \sqrt{u(x) + v(x)} dW, \tag{15}$$

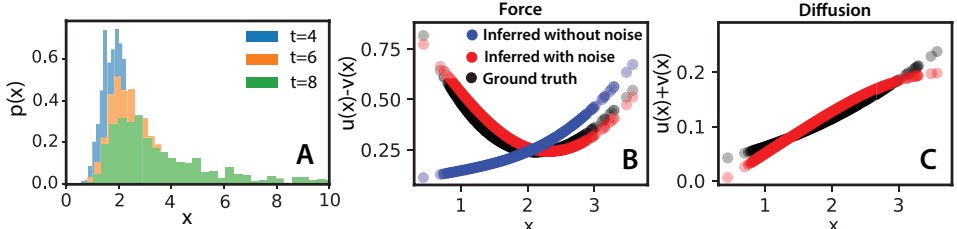

Figure 4: **Stochastic chemical kinetics.** (A) Cross-sectional empirical distributions are shown at times $t = \{4, 6, 8\}$. (B) Comparison between the force field jointly inferred with diffusion term (red) and without the noise model specified ($\mathbf{D} = 0$, blue). (C) Diffusion term jointly inferred with the force field.

where $W$ is a standard Wiener process and $V$ is the volume of the system. According to the law of mass-action applied to (14), the volumetric growth and decay rates of $X$ are, respectively, $u(x) = k_1 a x^2 + k_3 b$ and $v(x) = k_2 x^3 + k_4 x$. At this level of coarse-graining the fluctuations in the concentration of $X$ are directly prescribed by the volumetric rates $u$ and $v$. This observation is especially true in gene regulatory networks which are well described by high-dimensional CLEs [39], or simplifications of it [40].

In this example, we rely on the functional correspondence between the force field $(u - v)$ and diffusion $(u + v)$, to infer both. The approximate probability flow associated with (15) is given as

$$\frac{dx}{dt} = \left( \left[1 - \frac{\tilde{s}(x)}{2V}\right] u(x) - \left[1 + \frac{\tilde{s}(x)}{2V}\right] v(x) - \frac{1}{2V}\left[\partial_x u(x) + \partial_x v(x)\right] \right). \tag{16}$$

The simulation data is generated for the choice of parameters $k_1 = 0.3, k_2 = 0.02, k_3 = 1.2, k_4 = 1, a = 1, b = 1$, and $V = 20$ with $\delta t = 0.01$. We ran 1000 realizations of the process to generate $K = 20$ distributions separated by $\Delta t = 0.2$. The score was estimated via score matching (7) using a neural network with 2 hidden layers and 10 nodes per layer. The volumetric growth rate $u(x)$ and the decay rate $v(x)$ are modeled as the outputs of a single neural network with concentration $x$ as input, with 3 hidden layers and 10 nodes per layer. The neural network is then optimized based on the approximate probability flow (16) to solve the density fitting problem. In Fig. 4, we demonstrate that a few cross-sectional snapshots are sufficient to quantitatively infer both the force and diffusion field. We train our force model on data far from steady state and thus we only capture one stationary point in the force field as shown in Fig. 4B. We also provide a comparison with the TrajectoryNet [16]-like approach where the knowledge about the noise is ignored and, therefore, fails to identify the true force field. This clearly highlights the non-negligible role of multiplicative noise in chemical reaction networks, and how our inference framework can be used to distinguish the force field from intrinsic noise.

## 5  Discussion

In this paper, we developed a force-field inference approach for diffusion processes using cross-sectional samples given at a limited number of time points. We demonstrate our method on bio-physically relevant examples, and we show that our approach successfully identifies both linear and non-linear non-conservative force fields from non-stationary data, while it recovers the corresponding equilibrium dynamical model when applied to steady-state data. Additionally, in CLEs where fluctuations are prescribed by the underlying force field, we can jointly infer both the force field and the diffusion term. Unlike most trajectory inference methods, our approach allows simultaneously to (i) learn potentially nonlinear dynamical models consistent with cross-sectional measurements at all times and (ii) investigate the effect of arbitrary noise models, and in particular multiplicative models.

Future work should (i) investigate the effect of sampling rate and measurement noise on the accuracy of the inference scheme, (ii) extend the study on the joint inference of the force field and diffusion to realistic gene regulatory networks where noise is frequently ignored or assumed to be additive, conflicting with the ubiquitous presence of multiplicative noise in biological systems [2, 3, 20].

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
