# OpenReview forum: "Stochastic force inference via density estimation"
_NeurIPS.cc/2023/Workshop/AI4Science — NeurIPS2023-AI4Science Poster_

### Official Review · Reviewer_3nmy · 2023-10-22
**Missing context to related work. Still interesting and relevant for the workshop**

**Rating:** 6
**Confidence:** 4

**Review:**

### Summary

The authors use score matching together with simulation based training to infer a dynamical system’s forces. They provide experiments for equilibrium systems as well as non stationary processes where they show they can recover non-conservative forces.

### Strengths

1. The paper tackles an important real world task.
2. An interesting investigation of score matching for learning dynamics from observed data without trajectories.
3. Insights into how well simulation based training / how feasible it is to infer dynamics from crosssectional data
4. The dynamics inference problem is well explained and thoroughly motivated. The introduction is a valuable educational piece.
5. Novelty of inferring non-equilibrium dynamics.

### Content concerns

1. The paper claims the force inference to be a novelty of the work. However, “Two for One: Diffusion Models and Force Fields for Coarse-Grained Molecular Dynamics” already performs this task for real world systems by learning the forces of molecular dynamics from equilibrium data. This paper is not discussed or cited.
2. Force field inference via density fitting approach: this is an obvious way to learn the dynamics but it seems quite restrictive, expensive and infeasible for large systems. It seems that data at each timepoint is required to cover the whole space at that time. Integrating the ODE for training presumably is expensive and computing the divergence seems similarly problematic. This is not discussed anywhere.

### Presentation concerns

1.

### Recommendation

The paper provides interesting insights for dynamics inference and is a strong educational piece for the same. A large problem is the missing discussion of related work addressing a part of the problem which the paper claims as their novel contribution. I recommend acceptance

---

### Meta-Review · Area_Chair_Wtq4 · 2023-10-27

**Recommendation:** Accept (Poster)
**Confidence:** 3

**Metareview:**

The reviewers acknowledge the paper's novelty in applying score matching to learning dynamics and its relevance to the community. However, they raise concerns about the lack of discussion related to relevant works and fields. I strongly encourage the authors to incorporate these comments into the revision.